# Cutaneous Squamous Cell Carcinoma in Patients with Solid-Organ-Transplant-Associated Immunosuppression

**DOI:** 10.3390/cancers16173083

**Published:** 2024-09-04

**Authors:** Karam Khaddour, Naoka Murakami, Emily S. Ruiz, Ann W. Silk

**Affiliations:** 1Department of Medical Oncology, Dana-Farber Cancer Institute, Boston, MA 02215, USA; 2Center for Cutaneous Oncology, Dana-Farber Cancer Institute, Boston, MA 02115, USA; 3Harvard Medical School, Boston, MA 02115, USA; 4Division of Renal Medicine, Brigham and Women’s Hospital, Boston, MA 02115, USA; 5Department of Dermatology, Brigham and Women’s Hospital, Boston, MA 02115, USA

**Keywords:** solid-organ transplant, immunosuppression, immunocompromised, cutaneous squamous cell carcinoma, immunotherapy

## Abstract

**Simple Summary:**

Patients who have undergone solid-organ transplant are at higher risk of developing aggressive cutaneous squamous cell carcinoma (CSCC), which is associated with increased morbidity and mortality. There has been a major shift in the management landscape of locally advanced and metastatic CSCC with the introduction of immunotherapy. Despite this, the management of patients with a history of immunosuppression including solid-organ transplant recipients (SOTRs) remains challenging due to safety and efficacy concerns. This review addresses the unique aspects of biology and clinical care of this patient population and highlights recent advances.

**Abstract:**

The management of advanced cutaneous squamous cell carcinoma (CSCC) has been revolutionized by the introduction of immunotherapy. Yet, successful treatment with immunotherapy relies on an adequate antitumor immune response. Patients who are solid-organ transplant recipients (SOTRs) have a higher incidence of CSCC compared to the general population. This review discusses the current knowledge of epidemiology, pathophysiology, and management of patients with CSCC who are immunocompromised because of their chronic exposure to immunosuppressive medications to prevent allograft rejection. First, we discuss the prognostic impact of immunosuppression in patients with CSCC. Next, we review the risk of CSCC development in immunosuppressed patients due to SOT. In addition, we provide an overview of the biological immune disruption present in transplanted immunosuppressed CSCC patients. We discuss the available evidence on the use of immunotherapy and provide a framework for the management approach with SOTRs with CSCC. Finally, we discuss potential novel approaches that are being investigated for the management of immunosuppressed patients with CSCC.

## 1. Introduction

Cutaneous squamous cell carcinoma (CSCC) is a non-melanoma skin cancer and is the second most common cancer globally [1]. The estimated annual incidence is more than one million cases per year in the United States and is projected to increase [2]. Unlike melanoma, which has a high tendency for metastatic recurrence, most CSCC has excellent outcomes and are cured with surgery alone given the low frequency of metastatic disease (<5%) [3]. However, a subset of patients will develop locally unresectable or metastatic disease, eventually leading to high morbidity and mortality. CSCC is excluded from many national cancer registries due to its high incidence and the difficulty capturing multiple occurrences in the same patient, and thus, the precise disease-specific mortality is not available; however, it is estimated that about 1% of patients with CSCC will die from their disease. This translates to an annual mortality rate similar to that of melanoma [4]. Photodamage due to natural ultraviolet (UV) radiation from the sun poses the highest risk for the development of CSCC [5,6]. Damage from UV exposure can be exacerbated by medications that cause photosensitivity, such as azathioprine and voriconazole. Another important risk factor for CSCC is disruption of the host immune system leading to immunosuppression. Of importance, ultraviolet sun damage can induce immune cell exhaustion in skin tissue which could attenuate host defense mechanisms to prevent cancer development [7].

The introduction of immunotherapy in unresectable locally advanced, recurrent, and metastatic CSCC has transformed the management paradigm for immunocompetent patients. Clinical trials of immune checkpoint inhibitors (ICI) have demonstrated clinical efficacy, leading to improved objective response rates (ORRs) in locally advanced and metastatic CSCC. Two programmed death-1 (PD-1) inhibitors (cemiplimab and pembrolizumab) have been approved by the Food and Drug Administration (FDA) for unresectable locally advanced and metastatic CSCC based on an ORR ranging from 34% to 50% and a duration of response beyond 12 months in more than 80% of patients [8,9]. Long-term follow-up data show that 1 in 5 patients eventually achieve a complete response and the average duration of a response exceeds 3 years. Other ICIs have been investigated for CSCC. Nivolumab was associated with an ORR of 58.3% and a median overall survival (OS) of 20.7 months in an open-label, single-arm phase-II trial in patients with CSCC [10]. A recent clinical trial of a monoclonal antibody targeting programmed death ligand-1 (PD-L1) cosibelimab showed an ORR of 47.4% in patients with CSCC and is currently under review by the FDA [11]. Moreover, the use of anti-PD-1 cemiplimab in the neoadjuvant setting in resectable stage II-IV CSCC showed a complete pathological response rate of 51%, and the 12-month event-free survival was 89% in a phase-II single-arm trial and is currently being evaluated in a randomized trial to compare this approach with the standard of care (surgery with adjuvant radiation) [12,13].

Most clinical trials that have investigated ICI in solid cancers including CSCC have excluded patients who are immunocompromised, such as solid-organ transplant recipients (SOTRs), patients with hematopoietic stem cell transplants (HSCT), patients with autoimmune diseases, and patients with hematologic malignancies. Specifically, for SOTRs, there are safety concerns related to allograft rejection, leading to high morbidity and mortality. While immunosuppressed patients are considered to have a weakened antitumor immune response, recent real-world studies have shown comparable efficacy of immunotherapy in immunosuppressed individuals with CSCC compared to patients with intact immune systems [14,15].

A significant subset of patients with locally advanced or metastatic CSCC have a comorbid condition that is associated with immunosuppression. These conditions include the following: (1) solid-organ transplant recipients on immunosuppression, (2) hematologic malignancy, (3) chronic immunosuppression due to autoimmune disease and history of HIV infection, and (4) HSCT (Figure 1). The use of immunosuppressive medication such as calcineurin inhibitors, antimetabolites, and medication for HIV infection are associated with a higher risk of CSCC. Immune dysregulation can lead to decreased effector T-cell density and function and disruption of antigen-presenting mechanisms [14]. Suppressive cytokines in chronic inflammation and patients with a history of transplant and decreased permeability of effector immune cells into the tumor can promote CSCC progression [14]. This risk leading to CSCC development is largely due to extrinsic factors with the use of immunosuppressive medications such as chemotherapy or drugs used to prevent allograft rejection. This review discusses the current knowledge of epidemiology, pathophysiology, and management of immunosuppressed patients with CSCC with a focus on SOTRs.

## 2. Prognostic Impact of Immunosuppression on Patients with CSCC

Patients who are immunosuppressed due to their being SOTRs have an incidence rate of 812 CSCC cases per 100,000 person-years, which is higher than any other cancer [16]. The National Comprehensive Cancer Network (NCCN) guidelines consider immunosuppression as a stratification risk factor for identifying patients with high-risk CSCC. This is based on several studies showing higher rates of recurrence, metastases, and disease-specific death in immunosuppressed patients [17,18,19,20]. Primary CSCC lesions in immunosuppressed patients usually associate with aggressive histological features such as deep tissue infiltration, lymphovascular invasion, and perineural spread [21,22]. The impact of immunosuppression on clinical outcomes was observed in several studies. For example, a comparative study of 147 immunosuppressed and 649 immunocompetent patients found that immunosuppression was associated with a higher rate of disease-specific death in non-metastatic CSCC, where immunosuppression was defined as having history of SOT, hematologic malignancy, HSCT, active immunosuppressive therapy, HIV, and diabetes requiring insulin [23]. Similarly, other studies demonstrated higher rate of metastases in immunosuppressed patients [24,25,26]. A study of 849 immunosuppressed patients with CSCC in Europe found a trend towards higher rates of local recurrences and metastases, but this was not statistically significant [27].

In contrast, a few studies suggest that immunosuppression alone as a risk factor does not predict worse outcomes in patients with CSCC. This was highlighted in a study comparing 814 immunosuppressed CSCC patients to 4198 immunocompetent patients which demonstrated that the status of immunosuppression alone was not predictive of disease-specific death or metastases [28]. In clinical practice, however, patients with a history of immunosuppression and CSCC are considered to have high-risk disease given the higher complications that could be associated with cancer treatment.

## 3. Immunosuppression and Risk of CSCC

The studies that investigated the estimated risk of development of CSCC in SOTRs are listed in Table 1. Due to the extremely high risk of CSCC in SOTRs, an approach involving multidisciplinary discussion on patients who are SOTRs is crucial for reviewing if immunosuppressive medication can be modified and/or reduced to minimize the risk of CSCC development and progression.

The high incidence of CSCC in SOTRs was first observed in Australia in 1971 in kidney transplant recipients [29]. Subsequently, an Australian study of 1884 kidney allograft recipients found that skin cancers had the highest incidence among other cancers in post-transplant (77%), and this risk increased with time from transplant, suggesting an association between the duration of immunosuppression and development of CSCC [30]. Similarly, other studies confirmed that longer duration from time of transplant was associated with a higher risk of CSCC development with an increased incidence in one study from 7% at 1 year to 70% at 20 years post-transplant [31,32,33]. The association of CSCC development in SOTRs other than kidney transplant including heart, lung, liver, and pancreas transplant was documented in large cohort studies, all of which showed a significantly higher incidence rate compared to the general population [34,35,36,37]. Immunosuppressive medications used to prevent allograft rejection are the major contributing factor for the development of CSCC in SOTRs, and the risk is related to the class of medication [38]. Association between infection with the human papillomavirus and development of CSCC has been investigated in previous studies, but there has been no evidence to suggest a causality of virus-induced CSCC in SOTRs [39,40,41].

**Table 1 cancers-16-03083-t001:** Selected studies investigating the risk of CSCC incidence in SOTRs.

Author and Reference	Country	Number of Patients	Transplant Type	Immunosuppressive Drug	Risk of CSCC
Lindelof et al., 2000 [36]	Sweden	5356	Solid-organ transplant ^1^	Not reported	RR 108.6 in men and 92.8 in women compared to general population [S]
Jensen et al., 1999 [35]	Norway	2561	Kidney and heart transplant	Cyclosporine, azathioprine, prednisolone	65-fold increase in CSCC compared to the general population, and CSCC was higher in patients who received cyclosporine [S]
Sheil et al., 1977 [30]	Australia	1884	Kidney transplant	Not reported	The highest incidence of cancer post-transplant was skin cancer [N/A]
Bavinck et al., 1996 [32]	Australia	1098	Kidney transplant on cyclosporine, azathioprine, and/or prednisone	Cyclosporine, azathioprine, prednisolone	Cumulative incidence of NMSC increased from 7% at 1 year to 45% at 11 years [N/A]
Hartevelt et al., 1990 [31]	Netherland	764	Kidney transplant	Cyclosporine, azathioprine, prednisolone	A 253-time higher risk of CSCC compared to the general Dutch population [N/A]
Ramsay et al., 2007 [33]	UK	244	Kidney transplant	Not reported	Mean incidence per year of NMSC was 7.82%, and CSCC was the highest [N/A]
Dantal et al., 2018 [38] ^2^	Europe	120	Transplant recipients receiving calcineurin inhibitors were randomized to sirolimus vs. continuing calcineurin inhibitors ^2^	Sirolimus vs. CNI, MPA, azathioprine, prednisone	Secondary CSCC rate was 22% in the group that switched to sirolimus vs. 59% in the group that continued calcineurin inhibitors [S]
Ong et al., 1999 [34]	Australia	455	Heart transplant	Cyclosporine, azathioprine, prednisolone	Cumulative incidence of skin cancer was 31% and 43% at 5 and 10 years, respectively; CSCC to BCC ratio was 3:1 [S]
Brewer et al., 2009 [37]	US	312	Heart transplant	MTOR inhibitors, cyclosporine, MPA, CNI, azathioprine, prednisone	Cumulative incidence of NMSC was 20.4% and 46.4% at 5 and 15 years [N/A]

^1^ Included kidney, liver, heart, lung, liver, and pancreas transplants. ^2^ Randomized trial. Abbreviations: BCC: Basal cell carcinoma. CNI: Calcineurin inhibitor. MPA: Mycophenolic acid. CSCC: Cutaneous squamous cell carcinoma. NMSC: Non-melanoma skin cancer including CSCC and basal cell carcinoma. RR: Relative risk. S: Statistically significant. N/A: Statistical significance information not available or not applicable.

## 4. Classes of Immunosuppressive Medication

Long-term use of immunosuppressive medications is associated with a higher incidence and worse prognosis of CSCC. This association with skin cancer risk and prognosis seems to be medication-class-dependent. Immunosuppressive medications can affect tumor development and progression by various mechanisms including an acceleration of tumor growth and inhibition and tumor immunosurveillance. Host-intrinsic factors, such as concomitant immune dysregulation due to SOT, HSCT, and end-stage kidney diseases, as well as host-extrinsic factors such as the cumulative dose of immunosuppression medications [42,43], the class of medications, type of transplanted organ, degree of human leukocyte antigen (HLA) match with a donor are all crucial for understanding the increased risk of CSCC.

### 4.1. Calcineurin Inhibitors

Calcineurin inhibitors (CNIs, e.g., tacrolimus and cyclosporine) represent the mainstay for immunosuppressive regimens in SOT to prevent allograft rejection. CNIs inhibit the calcineurin-dependent, nuclear factor of the activated T-cell (NFAT) pathway, which regulates immune cell functions. Inhibition of this cascade in T-cells disrupts their activation and proliferation by downregulating the interlukin-2 (IL-2) signaling pathway [44]. Patients who have undergone solid-organ transplant and who are on CNIs have a higher incidence of CSCC and a rate of new secondary CSCC of 59% at 5 years [38,45]. The biological implications of tacrolimus use on CSCC development have been demonstrated in in vitro and in vivo studies, which found dose-dependent increase in proliferation, invasion, and cancer cell migration [46]. Several studies in patients treated with CNIs found no significant difference in the risk of CSCC development between tacrolimus and cyclosporine [47,48].

### 4.2. Antimetabolites

Antimetabolites that inhibit purine synthesis such as azathioprine (AZA) and mycophenolic acid (MPA) as well as antifolate drugs (methotrexate) inhibit DNA-damage repair, which could accelerate tumorigenesis by leaving ultraviolet-light-associated DNA damage unrepaired, eventually leading to CSCC development [49,50]. The use of AZA has been associated with a 3- to 8-fold increase in the risk of CSCC compared to the general population [51,52,53,54,55]. Interestingly, whole-exome sequencing of primary CSCC tumors demonstrated an azathioprine mutational signature suggesting a correlation between exposure to azathioprine and CSCC development and progression [56]. The risk of CSCC development in patients treated with MPA is lower than that with AZA [57]. A study in lung transplant patients found that switching immunosuppression from AZA to MMF led to a lower incidence of CSCC, suggesting that medication in the same class could lead to different risks for cancer development [58]. Similarly, methotrexate was found to be associated with an increased incidence of skin cancer in a case–control study in Europe, and this was dose-dependent [59].

### 4.3. Mammalian Target of Rapamycin (mTOR) Inhibitors

The class of mTOR inhibitors exert their immunosuppressive effects by inhibiting mTOR complexes, downstream of the phosphoinositide 3-kinase (PI3K)/protein kinase B (Akt) pathway, which in turn leads to the suppression of IL-2 and IL-15 cytokine-mediated T-cell proliferation. Sirolimus and everolimus are both used to prevent allograft rejection. A randomized clinical trial in post-kidney transplant patients comparing calcineurin inhibitor (cyclosporine) to mTOR inhibitor (sirolimus) found a lower incidence of recurrent CSCC in the group receiving sirolimus, and there was no difference in the graft rejection rate [60]. A follow-up randomized controlled study showed steady reduction in new CSCC in patients receiving sirolimus compared to cyclosporine (22% vs. 59%, respectively, *p* < 0.001) [38]. This accumulating evidence from randomized trials prompted the utilization of mTOR inhibitors, when possible, over other immunosuppressive drugs for patients at risk for CSCCs. However, the benefit of lower CSCC risk with the use of mTOR inhibitors must be balanced with its effectiveness to prevent allograft rejection, the increased risk of cardiovascular events, and the impairment of wound healing which could increase complications if surgical excision or radiation therapy (RT) are planned for the management of CSCC [61].

### 4.4. Corticosteroids

Corticosteroids are used in combinatorial regimens to prevent allograft rejection and GVHD. Some evidence suggests that a higher cumulative dose of corticosteroids is associated with a higher risk of CSCC compared to a low cumulative dose. However, these studies were confounded due to the use of other immunosuppressive drugs [53]. A systematic review of long-term use of systemic corticosteroids (defined as continuous use of corticosteroids for 30 days or more) found a slightly higher risk of cancers, but this did not include CSCC [62].

### 4.5. Novel Immunosuppressive Medication

Controversial evidence exists regarding a link between recently approved immunosuppressants such as Janus kinase (JAK) and Rho kinase (ROCK) inhibitors which are used for graft-versus-host disease (GVHD) prevention and higher incidence of aggressive CSCC. A study from the World Health Organization (WHO) database suggested an increased risk for CSCC with the use of JAK inhibitors [63]. In a post-marketing study of ruxolitinib in 3000 patients, only two patients were reported to develop SCC, although the study did not differentiate cutaneous from non-cutaneous SCC [64]. Similarly, a study of another JAK inhibitor (tofacitinib) showed no difference in CSCC [65]. Recently, our institution reported two patients who developed aggressive CSCC while on the JAK2 inhibitor ruxolitinib and the ROCK2 inhibitor belumosudil [66]. More research is needed on the risk of CSCC with JAK and ROCK inhibitors. Finally, belatacept, a fusion protein composed of an Fc fragment of IgG1 immunoglobulin linked to the extracellular domain of cytotoxic T-lymphocyte-associated antigen-4 (CTLA-4), is a newer immunosuppressive drug used to prevent allograft rejection in SOTRs. A small single-center study suggested that belatacept was associated with a lower incidence of CSCC compared to calcineurin inhibitors [67]. A more recent observational study found that belatacept was not associated with a lower risk of CSCC. Larger and longer follow-up data are necessary to fully assess the effect of belatacept on CSCC prevalence post-SOT [68].

## 5. Immune Disruption in the Tumor Microenvironment of CSCC in Immunosuppressed Patients

Effector cytotoxic T-cells play a central role in immune surveillance in the tumor microenvironment (TME). Alterations of the T-cell fate, as well as exhaustion, have been indicated to contribute to immune evasion, leading to cancer progression and decreased antitumor immunity [69]. Negative feedback interactions through highly expressed checkpoint receptors create an environment that allows cancer cell proliferation. In addition, other subsets of T-cells as well as antigen-presenting cells (APC), myeloid cells, and B-cells could disrupt the TME, leading to dysfunctional antitumor immunity [70].

Skin cancers, including CSCC, harbor one of the highest tumor mutational burdens (TMB) among all cancers [71]. This causes non-synonymous mutations and other genomic alterations leading to neoantigen formation that could be recognized by APCs through major histocompatibility complex (MHC) class I and II, eliciting a cascade of events that results in antitumor immunity. An effective response to such stimulation requires an intact immune cell function. In CSCC, the TME is characterized by immune disruption which promotes tumor growth and attenuates antitumor immunity. Both immunocompetent and immunosuppressed individuals with CSCC have CD3+ and CD8+ T-cells that are more abundant in their tumor tissue compared to normal skin tissue [72] (Figure 2). CSCC lesions from transplant patients are distinct due to the higher proportion of Foxp3+ T regulatory (Treg) cells to CD8+ T-cells, compared to tumors from immunocompetent individuals [72,73]. This in part could be mediated by increased migration of Tregs and homing through CCR5 (a chemokine receptor that regulates the trafficking of immune cells) and E-selectin (an endothelial-leukocyte adhesion molecule-2 which plays an important role in inflammation) [74,75]. The infiltration of Foxp3+ Treg cells has been shown to associate with decreased antitumor immunity [76]. In kidney transplant patients, the density of intratumoral effector immune CD8+ T-cells is decreased compared to immunocompetent patients, which is a negative prognostic marker in several solid cancers [77,78,79]. In addition, in a small study of CSCC from six transplant patients, T-cell clonality was decreased, suggesting a smaller array of effective immune cells [80].

Importantly, T-cells in CSCC from immunosuppressed patients have a similar exhaustion pattern compared to immunocompetent individuals. This suggests that the reversal of exhaustion in immunosuppressed patients with immunotherapy, such as the use of immune checkpoint inhibitors (ICIs), is feasible and could lead to the rejuvenation of antitumor immune response [80]. This is further highlighted by the evidence that checkpoint receptors such as PD-1/PD-L1 and lymphocyte activation gene-3 (LAG3) are present in CSCC [81,82]. Moreover, cytokines such as interferon gamma (INF-γ) are decreased in immunosuppressed patients due to attenuated T-cell response [83]. B-lymphocytes are also decreased in CSCC patients with a history of transplant, reflecting a diminished antitumor response in the TME [77,84,85]. The role of other immune cell subtypes such as myeloid suppressor cells has not been well characterized in CSCC patients, but in vivo mice studies found a dampening effect of tumor-associated neutrophils on antitumor immunity [86]. The effect of immunosuppressive medication and immune dysregulation in SOTRs can worsen the underlying defects in antitumor immunity in patients with CSCC. Specifically, the use of long and high doses of steroids which can induce apoptosis of lymphocytes including effector T-cells and can lead to upregulation of checkpoint receptors [87,88,89,90].

**Figure 2 cancers-16-03083-f002:**
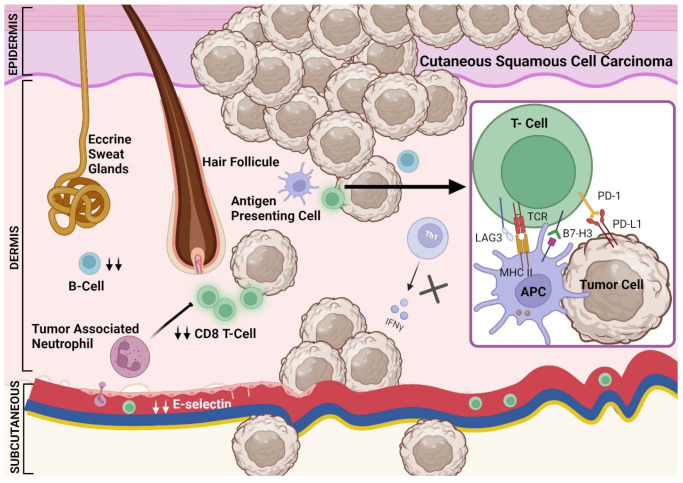
The tumor microenvironment in immunosuppressed patients with cutaneous squamous cell carcinoma. A dysregulated immune system in the TME is characterized by the presence of cytotoxic T-cells which are decreased compared to that in immunocompetent patients [79,80]. This in part could be mediated by decreased E-selectin [75]. In addition, CD4 T-cells are reduced in immunosuppressed patients leading to decreased INF-γ [83]. The density of other immune cells including B-cells is decreased in immunosuppressed patients [77,84]. Other mechanisms that could decrease the adaptive immune response in immunosuppressed patients include tumor-associated neutrophils [86]. The PD-1/PD-L1 expression is not altered in immunosuppressed patients with CSCC and B7H3 could be lower in immunosuppressed patients [81]. Other checkpoint receptors such as LAG3 could be also found in the TME of CSCC patients [82,85].

## 6. Current Treatment Options and Challenges

In unresectable, locally advanced, recurrent, and metastatic CSCC, the current guidelines support the use of cancer treatments such as RT and ICI as the first-line treatment in immunocompetent individuals. As most ICI clinical trials have excluded patients who are immunocompromised such as SOTRs, the safety and efficacy of these drugs had remained in question until recent prospective trials investigated different immunosuppressive regimens with ICI. Other systemic therapies such as chemotherapy and cetuximab have yielded suboptimal clinical outcomes in patients with CSCC and have not been formally investigated in SOTRs [91].

Evidence for conventional agents in the SOTR population is very slim and not encouraging. A case series of three patients with CSCC and a history of transplant (including heart, lung, and kidney) found that the use of oral capecitabine chemotherapy prevented new skin cancer development; however, this was limited to local CSCC and did not include patients with advanced-stage CSCC [92]. In addition, a death was reported in a lung transplant patient with CSCC who was treated with cetuximab and developed diffuse alveolar damage [93]. The poor efficacy of older systemic agents combined with safety concerns highlights the unmet need for better management approaches in this high-risk patient population. Due to the complexities of treating advanced CSCC in SOTRs, it is advantageous to make therapeutic decisions with multidisciplinary input.

### 6.1. Radiation Therapy for the Management of Unresectable Locoregional and Metastatic CSCC

Radiation therapy (RT) is a valuable option for definitive or palliative treatment of patients who are not candidates for surgery. The use of RT alone or in combination with systemic therapy (platinum-containing chemotherapy or cetuximab) as a definitive treatment approach in inoperable locoregional CSCC is associated with a high rate of local control and, given that more than 95% of CSCC recurrences are non-metastatic, may even improve survival [94]. Data from retrospective and prospective small cohorts demonstrated a local control rate ranging from 90% to 95% but with short duration of response and a 2-year disease-specific survival ranging between 50% and 85% [94,95,96]. Clinical outcomes did not differ according to the systemic therapy used with RT [94]. As such, the approach of concurrent RT with systemic therapy should be considered for patients with inoperable, non-metastatic CSCC, especially for SOTRs, for whom immunotherapy may be associated with high morbidity. In patients with metastatic disease, palliative RT alone or in combination with concurrent systemic therapy can achieve disease control [94,97]. The use of novel RT modalities including hypofractionated RT, stereotactic body radiation therapy, and brachytherapy have been described in CSCC and could be considered in patients who are poor candidates for systemic therapies [98,99,100,101,102,103,104].

### 6.2. Immune Checkpoint Inhibitors for the Treatment of Advanced CSCC

Given the significant impact of ICI on the field, it has become essential to consider an approach utilizing immunotherapy. However, using ICI in immunosuppressed patients is particularly challenging due to the efficacy and safety concerns, especially the risks of allograft rejection, as the immune checkpoint molecules are known to play critical roles in maintaining allograft tolerance.

Retrospective studies in real-world settings demonstrate lower efficacy of ICI in immunosuppressed patients and highlight the potential for severe adverse events such as allograft rejection and GVHD. For example, two systematic reviews of SOTRs with advanced cancers who were treated with ICI found that about 40% of patients experienced allograft rejection, of whom 70% experienced end-stage organ failure, and only a third had either complete or partial recovery of their allografts after rejection [105,106]. Interestingly, higher rates of rejection associated with anti-PD-1 compared to anti-CTLA-4 were observed in SOTRs and HSCT [106,107]. In kidney transplant recipients, similar rates of rejection can occur during treatment with ICI and could lead to irreversible allograft loss [108]. Another observation from the same study was that rates of rejection were lower in patients treated with mTOR inhibitors, suggesting that the choice of anti-rejection immunosuppression could be a modifiable risk factor for allograft rejection in patients treated with ICI. The use of ICI prior and after allogeneic HSCT led to antitumor efficacy comparable to the general population but was associated with higher morbidity due to significant acute and chronic rejection, some of which were fatal [109]. Combined, these studies suggest a comparable pattern of antitumor activity in cancer patients receiving ICI but provided evidence that adverse events related to allograft rejection are major safety concerns.

Although still limited, data are emerging on how to optimize the immunosuppressive regimen during ICI treatment to minimize rejection in SOTRs while maintaining antitumor efficacy. The alloreactive CD8+ T-cells that are enhanced by ICI and are responsible for rejection events might be dependent on the type, depth, and chronicity of immunosuppression [110,111]. Early case reports suggested that SOTRs could benefit from ICI, even while continuing to receive immunosuppressive medications, but treatment was sometimes followed by allograft rejection [112,113]. A seminal case report of a patient successfully treated with ICI drew attention due to the novel immunosuppressive regimen of an mTOR inhibitor and dynamic dosing of glucocorticoids [114].

To date, three prospective trials have reported outcomes on the safety and efficacy of ICI in kidney transplant recipients (KTRs) with CSCC and other cancers. The first prospective study to be reported was a phase I multisite Australian trial of nivolumab in KTR with various metastatic solid cancers (N = 17) [115]. They maintained patients on their pre-trial immunosuppressive regimen during nivolumab treatment. Most patients were on a two-drug immunosuppressive regimen (53%) or a three-drug immunosuppressive regimen (41%), and the most frequently used drug was prednisone (76%). The investigators reported no irreversible allograft rejection in 17 patients (100%) who were followed for a median of 28 months and were treated with a median of three infusions with an anti-PD-1 [115]. None of the six patients with CSCC responded. Another prospective trial evaluated a standardized regimen of low-dose tacrolimus plus low-dose prednisone during treatment with nivolumab in eight KTRs with advanced melanoma and non-melanoma skin cancers [116]. The primary composite endpoint was overall response rate without allograft loss at 4 months. None of the eight evaluable patients responded, and one patient experienced treatment-related allograft loss (TRAL). Patients who experienced progression were allowed to escalate therapy, and among six patients who received combined treatment with ipilimumab plus nivolumab, there were two complete responses (one with TRAL) and four with progressive disease (one with TRAL). The analysis of PD-L1 and rejected allografts from this trial demonstrated high PD-L1 expression as well as activated T-cells, suggesting the importance of the PD-1/PD-L1 pathway and T-cell-mediated response in allograft rejection [116]. The final study (CONTRAC) was conducted at our institution in 12 KTRs with advanced CSCC to evaluate the safety and efficacy of cemiplimab (anti-PD-1) in conjunction with a standardized regimen for immunosuppression (the Harvard regimen). The Harvard regimen consists of a dynamic schedule of moderate doses of prednisone (mini-pulse dose of 40 mg daily for 4 days starting the day before each infusion, followed by 20 mg daily for three days, then 10 mg daily) plus continuous use of an mTOR inhibitor at therapeutic levels (goal trough 4–6 ng/mL). The primary endpoint was safety, and secondary endpoints included ORR [117]. At a median follow up of 6.8 months, there were no episodes of rejection or kidney allograft loss, and the ORR was 46% in 11 evaluable patients. These trials addressed an unmet need and demonstrated proof of concept for ICI in the management of CSCC in KTR.

Despite the advances in our management approach using immunotherapy, several challenges exist regarding the safety and efficacy of implementing a similar strategy in SOTRs other than KTRs. For KTRs, we are fortunate to have renal replacement therapy as a fallback in the case of allograft rejection and failure. Replacement therapy is not available for other organs, such as the liver, heart and lungs, which makes ICI therapy more risky in these patients [118]. At our institution, we follow a treatment algorithm that involves a multidisciplinary approach to medical decision-making regarding modulation of immunosuppression and treatment selection (Figure 3). If a patient is not considered to be a candidate for surgery or radiation therapy, systemic options are discussed. For KTRs with CSCC in need of systemic therapy, after appropriate discussions of risks and benefits and recognizing that the clinical data are very limited in this population, we offer ICI with the Harvard regimen [117]. We avoid using continuous high-dose prednisone as this has been shown to dampen the antitumor efficacy of ICI in solid cancers [117,119,120].

**Figure 3 cancers-16-03083-f003:**
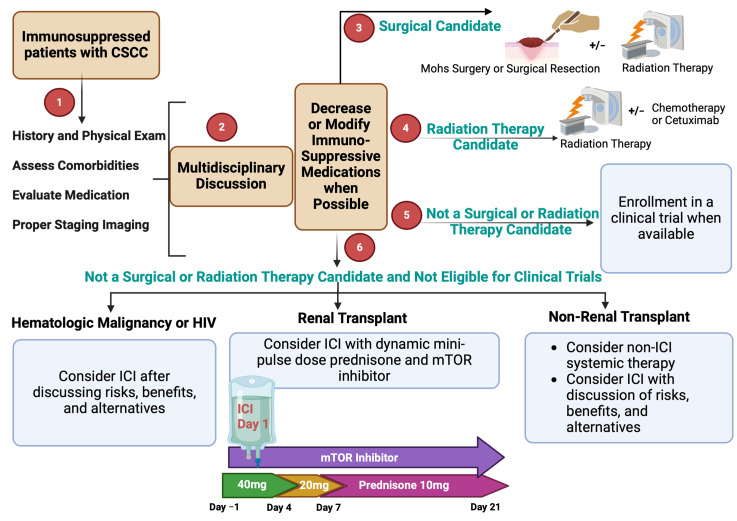
Approach to the evaluation and management of immunosuppressed patients with CSCC.

### 6.3. Monitoring for Rejection during Treatment with ICI

Given that rejection remains a major safety concern during treatment with ICI, close monitoring is essential for assessing the risks and benefits of continuing immunotherapy. Most episodes of rejection during treatment with ICI occur between 1.7 and 5 weeks of initiating treatment [118]. The majority of rejection episodes reported in the literature were T-cell-mediated (60%) and about 40% were mixed T-cell- and antibody-mediated [105], and as such, involving a transplant specialist is crucial when considering immunotherapy in an SOTR. Laboratory monitoring in the recently published protocols for KTRs included monitoring the basic metabolic panel (BMP) and urine protein/creatinine ratio. In our practice, we perform these labs weekly for the first 8–12 weeks and then with every immunotherapy infusion. Novel technology using donor-derived cell-free DNA (dd-cfDNA) blood testing has been developed in the post-transplant setting which is increasingly used to monitor for allograft rejection as an alternative to allograft biopsies. This quantitative assay can provide information on the fraction of unique donor single-nucleotide polymorphisms that are shed from dead allograft cells compared to the total DNA in the sample [121,122]. The use of dd-cfDNA may be helpful as an adjunct non-invasive biomarker during the treatment of kidney, heart, and lung transplant recipients to monitor for allograft rejection during ICI therapy [116,117].

## 7. Immunotherapy in CSCC Patients with Other Forms of Immunosuppression

There are few data regarding response patterns to ICI in immunosuppressed patients with CSCC due to concomitant hematologic malignancy or a history of HSCT. Retrospective studies and case reports have suggested an efficacy comparable to that seen in clinical trials in patients with CLL, myelodysplastic syndrome (MDS), lymphoma, and other hematologic malignancies but larger prospective studies are essential to confirm these findings [10,123,124,125].

## 8. Novel Treatment Approaches for CSCC Patients with Immunosuppression

Several studies are ongoing and in planning to assess the safety and efficacy of immunotherapy in immunosuppressed patients with CSCC (Table 2). A currently accruing trial will evaluate the safety and efficacy of ipilimumab and nivolumab with sirolimus and prednisone in kidney transplant patients with cutaneous cancers (NCT05896839). Another ongoing trial is evaluating intratumoral injection of an engineered oncolytic herpes simplex virus type-1 (RP1) that expresses granulocyte-macrophage colony-stimulating factor and a fusogenic glycoprotein in patients with a history of allogeneic transplant and skin cancers including CSCC (NCT04349436). Preliminary results from Part A of the trial, which included KTRs, demonstrated 27% ORR in 11 evaluable patients with an acceptable safety profile and no allograft rejection. Results from Part B that includes other allogeneic transplant recipients have not been reported yet [126].

**Table 2 cancers-16-03083-t002:** Selected ongoing trials of immunotherapy in SOTRs with skin cancers.

Study Design	Patient Population	Intervention	Results	NCT Number
Phase-I/II multi-institutional	Skin cancers with kidney transplant	Sirolimus + Prednisone + Ipilimumab and Nivolumab	No results available	NCT05896839
Phase IB/II multi-institutional [126]	Part A: kidney transplant Part B: any allogeneic transplant	RP1 intratumoral injection	Interim results from Part A: 27% ORR in 11 evaluable patients, 0 rejection	NCT04349436

## 9. Conclusions

SOTRs are at an increased risk for developing multiple CSCCs, some of which demonstrate aggressive behavior and lead to morbidity and mortality. Immunosuppression medications should be reviewed at the time of diagnosis of CSCC. Working in collaboration with the transplant specialist team, the immunosuppression regimen should be modified and/or reduced. Across the literature, there appears to be class differences in the risk of developing CSCCs; most of the data suggest that the incidence rates with azathioprine and calcineurin inhibitors is greater than those with mTOR inhibitors. There also may be important within-class differences, such as among the antimetabolites, as AZA may promote the development of skin cancers more than MPA. Historically, ICIs were contraindicated in advanced CSCC patients requiring systemic therapies. However, recent prospective trials in KTRs have demonstrated the safety and efficacy of ICI with a standardized immunosuppressive regimen, particularly the Harvard regimen consisting of mini-pulse doses of prednisone plus mTOR inhibitor. Despite this progress, the implementation of a similar approach in patients with other solid-organ transplants, such as liver, heart, and lung transplants, remains challenging, given the concern for high morbidity and mortality and the inability to modify immunosuppressive regimens due to higher risk of allograft rejection. As such, further prospective trials using novel immunotherapy and rationally selected immunosuppressive regimens are needed to address the unmet gap in this patient population.

## Figures and Tables

**Figure 1 cancers-16-03083-f001:**
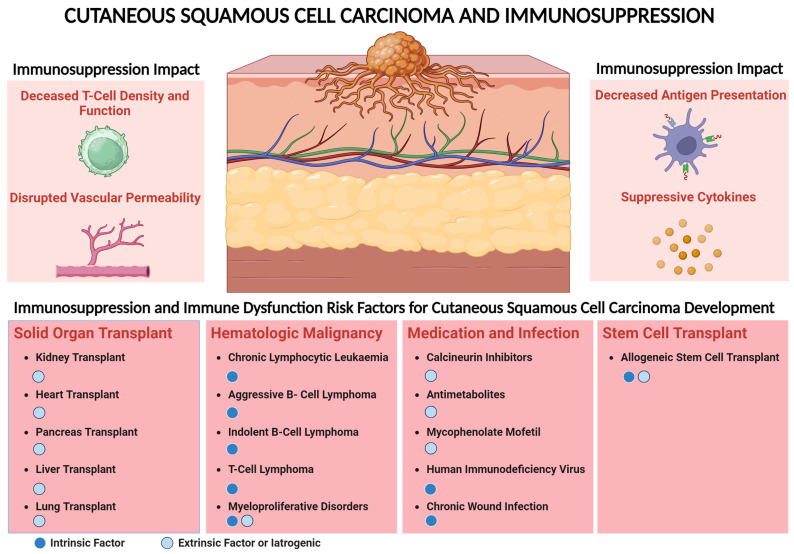
Immunosuppression impact and risk factors associated with cutaneous squamous cell carcinoma (CSCC). Use of immunosuppressive medication such as calcineurin inhibitors, antimetabolites, and HIV infection are associated with higher risk of CSCC. Immune dysregulation can lead to decreased effector T-cell density and function and disruption of antigen-presenting mechanisms. Suppressive cytokines in chronic inflammation and patients with a history of transplant and decreased permeability of effector immune cells into the tumor can promote CSCC progression. This risk leading to CSCC development could be either intrinsic, e.g., because of modulation of the host immune system due to transplant or hematologic malignancy targeting specific immune cells, or due to extrinsic factors, e.g., the use of immunosuppressive medication including patients with transplant to prevent allograft rejection.

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
