# Peer review of "Cutaneous Squamous Cell Carcinoma in Patients with Solid-Organ-Transplant-Associated Immunosuppression"

_cancers, 2024, doi:10.3390/cancers16173083_

Round 1

Reviewer 1 Report

Comments and Suggestions for Authors

Dear authors,

I offer my congratulations for the high quality of your paper. It's well-conceived and -organized. However, I would underscore one weakness: you discussed how difficult the management of CSCC patients being also SOTR is, mainly because of negative interactions of ICIs with immune suppressive therapies administered for avoiding allograft rejection. Your target is represented by that population with locally advanced or metastatic CSCC not eligible for surgery. The latter impediment does not automatically imply that the patient will also be not eligible for radiotherapy, whose benefit in such scenarios is undoubted. In your exhaustive review, radiation therapy is mentioned in passing only two times (lines 62 and 368). In this manner, the therapeutic potential of radiotherapy in this patient setting comes out as largely underrated and I don't agree with your choice of omitting its discussion. Radiotherapy, thanks to the more recent technological advances, has no particular risks even in a fragile population like the one investigated by you, and can be delivered both curatively and palliatively, achieving very satisfying results. For example, there are several series reporting an ORR significantly higher than that with ICIs (which often does not go beyond 50%...) and durable local control, which is the main outcome in a histology cancer with a poor metastatic potential (between 0.4% and 2%). However, as reviewed in PMID: 35454779, radiotherapy can improve survival even in the metastatic stage. New radiotherapy techniques are promising in bulky metastases, as illustrated in PMID: 35280772, and potentially able to elicit an endogenous antitumor immune response non-crossreacting with the allograft. Therefore, I think it's a pity that you totally missed radiotherapy in the management approach for CSCC patients with SOTR. Such a condition is not a contraindication to the use of radiotherapy and ICIs should be used upon its failure and no room for further irradiation. For the above reasons, I strongly suggest that you add at least one paragraph on the abilities of radiotherapy. Within it, you can discuss all the following references: PMID: 35280772, PMID: 35454779, PMID: 27908679, PMID: 37179219PMID: 30521102, PMID: 29524319, PMID: 28843727, PMID: 34648871 and PMID: 33948304. With these 9 references, you should be able to prepare an insightful and thorough section. Please, discuss them all.

Minor comments:

Line 51: "a median duration of response of more than 80% beyond 12 months", measure unit of DOR is time, the 80% value is inappropriate, as well as "median" in relation to 80%. Maybe, you should rephrase it as "a duration of response beyond 12 months in more than 80% of patients". Please, check it and clarify.

In the caption of Figure 1, you listed the "modulation of the host immune system due to transplant" as an intrinsic factor. Why? Such a modulation was extrinsic to avoid allograft rejection. Are you referring to GVHD? Please, clarify it, as well as the reason why all the transplants in the pink box are also intrinsic factors other than extrinsic.

In line 91, there are four 0 after the comma.

In line 117, to make sense, maybe it is necessary to replace "CSCC" with "SOTR".

In line 148, explain what the tumor immunotoxicity is.

In line 184, "medicated" should be corrected with "mediated".

The sentence in line 220 is unclear. What does "IO-responsive mean? What is counterintuitive?

In line 266, "can exacerbate the underlying attenuated anti-tumor immunity" could be re-phrased as "can increase the underlying attenuation of anti-tumor immunity", if this is what you mean.

Lines 282 and 283 are not true. Current guidelines also support the use of RT as a definitive therapy in unresectable, locally advanced, recurrent CSCC. Please, see my first major comment.

In Figure 3 RT exists only beside (+ or -) surgery (Should you have meant RT only as an adjuvant therapy?). Indeed, it also exists independently from surgery and it can be definitively curative.

Author Response

Reviewer

I offer my congratulations for the high quality of your paper. It's well-conceived and -organized. However, I would underscore one weakness: you discussed how difficult the management of CSCC patients being also SOTR is, mainly because of negative interactions of ICIs with immune suppressive therapies administered for avoiding allograft rejection. Your target is represented by that population with locally advanced or metastatic CSCC not eligible for surgery. The latter impediment does not automatically imply that the patient will also be not eligible for radiotherapy, whose benefit in such scenarios is undoubted. In your exhaustive review, radiation therapy is mentioned in passing only two times (lines 62 and 368). In this manner, the therapeutic potential of radiotherapy in this patient setting comes out as largely underrated and I don't agree with your choice of omitting its discussion. Radiotherapy, thanks to the more recent technological advances, has no particular risks even in a fragile population like the one investigated by you, and can be delivered both curatively and palliatively, achieving very satisfying results. For example, there are several series reporting an ORR significantly higher than that with ICIs (which often does not go beyond 50%...) and durable local control, which is the main outcome in a histology cancer with a poor metastatic potential (between 0.4% and 2%). However, as reviewed in PMID: 35454779, radiotherapy can improve survival even in the metastatic stage. New radiotherapy techniques are promising in bulky metastases, as illustrated in PMID: 35280772, and potentially able to elicit an endogenous antitumor immune response non-crossreacting with the allograft. Therefore, I think it's a pity that you totally missed radiotherapy in the management approach for CSCC patients with SOTR. Such a condition is not a contraindication to the use of radiotherapy and ICIs should be used upon its failure and no room for further irradiation. For the above reasons, I strongly suggest that you add at least one paragraph on the abilities of radiotherapy. Within it, you can discuss all the following references: PMID: 35280772, PMID: 35454779, PMID: 27908679, PMID: 37179219, PMID: 30521102, PMID: 29524319, PMID: 28843727, PMID: 34648871 and PMID: 33948304. With these 9 references, you should be able to prepare an insightful and thorough section. Please, discuss them all.

Response to Reviewer: We thank the reviewer for recognizing this omission. We certainly agree that RT is an important treatment option for the management of CSCC. To address this omission, we added a new sub-section (section 6.1, now located at lines 326-342 on the tracked version) to highlight the role of RT in the management of CSCC as a single treatment modality or in combination with non-immunotherapy treatment options. We appreciate the articles that were suggested by the reviewer and added them as new references.  

Minor comments:

Line 51: "a median duration of response of more than 80% beyond 12 months", measure unit of DOR is time, the 80% value is inappropriate, as well as "median" in relation to 80%. Maybe, you should rephrase it as "a duration of response beyond 12 months in more than 80% of patients". Please, check it and clarify.

Response to Reviewer: We have corrected this error (line 60). 

In the caption of Figure 1, you listed the "modulation of the host immune system due to transplant" as an intrinsic factor. Why? Such a modulation was extrinsic to avoid allograft rejection. Are you referring to GVHD? Please, clarify it, as well as the reason why all the transplants in the pink box are also intrinsic factors other than extrinsic.

Response to Reviewer: Although there is translational evidence that there is a component of intrinsic immunosuppression caused by the graft itself (PMID 30936086, 36906586, 38498808), that literature is outside the scope of this clinical review. We agree with the reviewer that the risk of immunosuppression in solid organ transplant recipients in mainly due to extrinsic factors. To reflect this accurately in the figure, we have removed the dark blue circles in the figure. We also explain this in the figure to make it easier to read and understand.

In line 91, there are four 0 after the comma.

Response to Reviewer: Corrected.

In line 117, to make sense, maybe it is necessary to replace "CSCC" with "SOTR".

Response to Reviewer: Corrected.

In line 148, explain what the tumor immunotoxicity is.

Response to Reviewer: This was a typo and it was deleted.

In line 184, "medicated" should be corrected with "mediated".

Response to Reviewer: This was corrected.

The sentence in line 220 is unclear. What does "IO-responsive mean? What is counterintuitive? Response to Reviewer: We thank the reviewer for the comment. The mechanism of action of belatacept is through the inhibition of T cell stimulation through binding to CTLA-4 which is a checkpoint protein that regulates immunosurveillance and can also be targeted with cancer directed immunotherapy such as ipilimumab, which is approved to treat advanced melanoma. To avoid confusion and distracting information, we deleted the sentence “However, given IO responsiveness ....” (line 245)

In line 266, "can exacerbate the underlying attenuated anti-tumor immunity" could be re-phrased as "can increase the underlying attenuation of anti-tumor immunity", if this is what you mean.

Response to Reviewer: We thank the reviewer for the comment. The sentence was edited according to the reviewer’s suggestion (line 290-291).

Lines 282 and 283 are not true. Current guidelines also support the use of RT as a definitive therapy in unresectable, locally advanced, recurrent CSCC. Please, see my first major comment.

Response to Reviewer: We thank the reviewer for the comment. The sentence was corrected to reflect that RT represents a valid option supported by guidelines for this patient population. (now located at line 309)

In Figure 3 RT exists only beside (+ or -) surgery (Should you have meant RT only as an adjuvant therapy?). Indeed, it also exists independently from surgery and it can be definitively curative.

Response to Reviewer: We thank the reviewer for the comment. The figure has been edited to reflect that RT is an option as a single treatment modality.

Reviewer 2 Report

Comments and Suggestions for Authors

The use of immunosuppressive drugs, although essential in many clinical situations, carries the risk of adverse effects. Some side effects of these drugs are directly or indirectly related to the pharmacological properties, including an increased risk of severe infections and the development of cancers. In this paper, the Authors analyzed the occurrence of cutaneous squamous cell carcinoma in patients with solid organ transplant-associated immunosuppression. Overall, I find the work valuable and interesting. The paper includes a theoretical introduction on the occurrence and treatment of cutaneous squamous cell carcinoma, as well as an analysis of immunosuppressants in the context of prognostic factors and the risk of CSCC for different groups of the drugs. As part of the topic analysis, the Authors also described the role of the tumor microenvironment and the issues related to current therapeutic solutions. However, before deciding to publish the work, please consider two issues:

  1. If possible, to increase the clarity of the presented information, please add a column referring to the drugs analyzed in the study.
  2. One of the main risk factors for developing CSCC is excessive sun exposure, especially in relation to the ultraviolet spectrum. This extremely important and fundamental aspect has been completely overlooked in this paper. I suggest linking the changes associated with photo-damage to skin cells with the immunosuppressive action. Additionally, it seems necessary to consider the issue of the phototoxicity of immunosuppressive drugs.

Author Response

The use of immunosuppressive drugs, although essential in many clinical situations, carries the risk of adverse effects. Some side effects of these drugs are directly or indirectly related to the pharmacological properties, including an increased risk of severe infections and the development of cancers. In this paper, the Authors analyzed the occurrence of cutaneous squamous cell carcinoma in patients with solid organ transplant-associated immunosuppression. Overall, I find the work valuable and interesting. The paper includes a theoretical introduction on the occurrence and treatment of cutaneous squamous cell carcinoma, as well as an analysis of immunosuppressants in the context of prognostic factors and the risk of CSCC for different groups of the drugs. As part of the topic analysis, the Authors also described the role of the tumor microenvironment and the issues related to current therapeutic solutions. However, before deciding to publish the work, please consider two issues:

  1. If possible, to increase the clarity of the presented information, please add a column referring to the drugs analyzed in the study.

Response to Reviewer: We thank the reviewer for the comment. The table has been edited and we added a column of the immunosuppressive drugs that were included in the referenced studies which is included in Table 1 (located at line 160 on the tracked version).

  1. One of the main risk factors for developing CSCC is excessive sun exposure, especially in relation to the ultraviolet spectrum. This extremely important and fundamental aspect has been completely overlooked in this paper. I suggest linking the changes associated with photo-damage to skin cells with the immunosuppressive action. Additionally, it seems necessary to consider the issue of the phototoxicity of immunosuppressive drugs.

Response to Reviewer: We thank the reviewer for the comment. We recognize the importance of highlighting the role of sun damage and phototoxic medications on the development of CSCC. We have added this to the manuscript which can be found in lines 46-52.

Reviewer 3 Report

Comments and Suggestions for Authors

Dear authors,

greetings for your review manuscript. It is very interesting and detailed for the field. Figures are good and illustrative. References are appropriate and updated. 

I suggest only minor revision for publish. 

Because of this manuscript is a review I suggest you to specify in the text that CSCC is a non-melanoma skin cancers and (correctly by citing reference 1 as you write) describe why its differs from cutaneous melanoma.

LINE 39-40 : “CSCC is excluded from many national cancer registries": explain the reason why, please.

LINE 72-76: Add to text something what you breafly describe in figure 1 legend, please. And / or detail with references.

FIGURE 1 : Editing error at the end of legend : a double end point..

LINE 163-164 : I suggest to correct "migratory capabilities "with " cancer cell migration".

LINE 184 editing error "mediated" .

LINE 195: If you are agree I think you have to explain something about wound healing because it refers to cancer cells characteristic in a phrase generally contextualized. Furthermore I do not find this description in reference 57are you sure this is the correct reference for this methods/cellular mechanism?

LINE 240: “in their tumor” : means in tissue histology samples or circulating cells ? Specify, please.

LINE 320: editing error “seminal “.

LINE 373 : editing error , end poin missing .

LINE 396 : correct with “ there are few data regarding….”, please.

LINE 545 : editing error at the end of authors name.

I have a question about phatologycal diagnosis: Why don 't you added a reference about these CSCC details? For example Alam et al. 2018 Or another one more recent.

Thank you and greetings.

Author Response

Dear authors,

greetings for your review manuscript. It is very interesting and detailed for the field. Figures are good and illustrative. References are appropriate and updated.

I suggest only minor revision for publish.

Because of this manuscript is a review I suggest you to specify in the text that CSCC is a non-melanoma skin cancers and (correctly by citing reference 1 as you write) describe why its differs from cutaneous melanoma.

Response to reviewer: We thank the reviewer for their comment. We have added the suggested edits which can be found in lines 35-39.

LINE 39-40 : “CSCC is excluded from many national cancer registries": explain the reason why, please.

Response to reviewer: We thank the reviewer for the comment. The reason from the inaccurate capture of CSCC in cancer registries is due to the high frequency of its incidence and the difficulty capturing the disease which can occur multiple times in the same patient. This explanation was added to lines 42-43.

LINE 72-76: Add to text something what you briefly describe in figure 1 legend, please. And / or detail with references.

Response to reviewer: We thank the reviewer for their comment. We have added a paragraph explaining what is mentioned in figure 1 legend. This can be found in lines 86-95

FIGURE 1 : Editing error at the end of legend : a double end point..

Response to reviewer: We thank the reviewer for the comment. This was corrected.

LINE 163-164 : I suggest to correct "migratory capabilities "with " cancer cell migration".

Response to reviewer: We thank the reviewer for the comment. This was corrected.

LINE 184 editing error "mediated" .

Response to reviewer: We thank the reviewer for the comment. This was corrected.

LINE 195: If you are agree I think you have to explain something about wound healing because it refers to cancer cells characteristic in a phrase generally contextualized. Furthermore I do not find this description in reference 57: are you sure this is the correct reference for this methods/cellular mechanism?

Response to reviewer: We thank the reviewer for the comment. The paragraph in the referenced section discusses some of the disadvantages associated with the use of mTOR inhibitors. One of the complications that can happen with mTOR inhibitors include impaired wound healing which could complicate the care of patients with CSCC that requires surgical resection or RT which could lead to formation of wounds during treatment.

LINE 240: “in their tumor” : means in tissue histology samples or circulating cells ? Specify, please.

Response to reviewer: We thank the reviewer for the comment. This refers to tumor tissue. We have added that to the manuscript to clarify that (now located at line 266).

LINE 320: editing error “seminal “.

Response to reviewer: We thank the reviewer for the comment. We have corrected this (now located at line 372).

LINE 373 : editing error , end point missing .

Response to reviewer: We thank the reviewer for the comment. We have corrected this.

LINE 396 : correct with “ there are few data regarding….”, please.

Response to reviewer: We thank the reviewer for the comment. We have corrected this (no located at line 442).

I have a question about pathological diagnosis: Why don 't you added a reference about these CSCC details? For example Alam et al. 2018 Or another one more recent.

Response to reviewer: We thank the reviewer for the comment. We have added the reference by Alam et al to the introduction (reference 3).

Round 2

Reviewer 1 Report

Comments and Suggestions for Authors

The paper can be pusblished in its current version.

Reviewer 2 Report

Comments and Suggestions for Authors

The manuscript has been revised by the Authors according to the suggestions. I believe the paper can be published in its current form.